# Development of Lab-on-a-Chip LAMP and Real-Time PCR Assays to Detect Aflatoxigenic *Aspergillus flavus* and *Aspergillus parasiticus* in Hazelnuts

**DOI:** 10.3390/toxins17100510

**Published:** 2025-10-17

**Authors:** Slavica Matić, Livio Cognolato, Martina Sanna, Monica Mezzalama, Riccardo Laurenti, Davide Spadaro

**Affiliations:** 1Interdepartmental Centre for the Innovation in the Agro-Environmental Sector (AGROINNOVA), University of Torino, 10095 Grugliasco, Italy; slavica.matic@unipa.it (S.M.); martina.sanna@unito.it (M.S.); monica.mezzalama@unito.it (M.M.); 2Department of Agricultural, Forest and Food Sciences, University of Torino, 10095 Grugliasco, Italy; 3Department of Agricultural, Food and Forest Sciences (SAAF), University of Palermo, Viale delle Scienze, 90128 Palermo, Italy; 4LAMP s.r.l., 10010 Scarmagno, Italy; livio.cognolato@pro-jet-consulting.eu (L.C.); riccardo.laurenti@lampgroup.it (R.L.)

**Keywords:** isothermal amplification, LoC assay, microfluidic device, aflatoxins, food safety, mycotoxins

## Abstract

Aflatoxins, which are potentially genotoxic and carcinogenic substances, are mainly produced by the *Aspergillus* section *Flavi*, including *Aspergillus flavus* and *A. parasiticus*. Current *Aspergillus* spp. detection is often based on molecular methods, such as real-time PCR and loop-mediated isothermal amplification (LAMP), targeting genes of the aflatoxin biosynthetic cluster. In this study, we developed a Lab-on-a-Chip (LoC) method based on real-time PCR and on LAMP for the specific detection of aflatoxigenic strains of *A. flavus* and *A. parasiticus* from infected hazelnuts. LoC-LAMP and LoC-real-time PCR assays were tested in terms of specificity, sensitivity, speed, and repeatability. The microfluidic chip allowed quick, specific, sensitive, simple, automatized, cheap, and user-friendly detection of aflatoxigenic strains of *A. flavus* and *A. parasiticus*. The LoC-LAMP showed a limit of detection (LOD) of 10 fg of DNA, while the LoC-real-time PCR showed a LOD of 10 pg of DNA. Achieving comparable sensitivity to that of LAMP and real-time PCR techniques, both LoC methods developed in this work offer the advantages of automation, minimal sample requirements, reagent requirements, and cost-effectiveness. Overall, the developed methods open the perspective for alternative monitoring of aflatoxigenic fungi in the agri-food industry.

## 1. Introduction

Food contamination with pathogenic microorganisms and viruses, as well as their activity, growth, persistence, and toxin production in food, poses serious public health risks [1,2]. Most foodborne pathogens are bacteria, followed by fungi, which have been reported both in raw agricultural and processed food products [3,4,5]. Food quality control systems, which monitor raw materials, semi-finished products, and final food products, are therefore of crucial importance. The physical and chemical properties of the raw materials generally depend on the temperature, pressure, and humidity associated with the harvest and storage conditions, which may considerably influence the development of food contaminants [6]. The low sugar and high oil content in nuts seems to favor aflatoxin production, predominantly produced by *Aspergillus flavus* Link [7].

*A. flavus* belongs to *Aspergillus* section *Flavi*, which contains over 30 species, including *A. parasiticus* Speare and *A. toxicarius* Murak. Most species are able to produce aflatoxins B1 (AFB1), B2 (AFB2), G1 (AFG1), and G2 (AFG2) [8]. In particular, *A. flavus* and *A. parasiticus* may produce aflatoxins in nuts during the cropping season in the field and during the harvest operations and post-harvest storage period, and their production is considerably affected by humidity and temperature conditions [9,10]. Besides aflatoxigenic species, there are also non-aflatoxin species, such as *A. oryzae* and *A. sojae*, which play an important role in producing fermented foods used in East and Southeast Asia [11]. The occurrence of aflatoxigenic fungi is strongly influenced by environmental conditions, particularly temperature and humidity; therefore, the occurrence of aflatoxins can be affected by climate change, particularly by global warming [12,13].

Food products are complex matrices, and their samples are chemical mixtures of heterogeneous molecules (mainly carbohydrates, proteins, and lipids) in a complex physical matrix (amorphous solids, liquids, gels, macro-organelles, cells, macromolecules, and crystals), which makes the monitoring of food processes demanding, in terms of time and sampling [14]. It is therefore evident that an effective strategy should analyze not only the process inputs and outputs, but it should also continuously evaluate the process by automatic real-time control. Current molecular diagnostic techniques for the detection of *A. flavus* and *A. parasiticus* are based on conventional PCR, real-time PCR, and loop-mediated isothermal amplification (LAMP) [15,16,17]. These techniques permit the differentiation between aflatoxin-producing and non-aflatoxin-producing strains of *Aspergillus* section *Flavi*. Furthermore, there are serological methods for total or partial aflatoxin detection in food matrices [18].

Automation and miniaturization of analytical procedures provide many benefits in diagnostics due to limited sample and reagent quantities, test speed, and sensitivity and minimum sample handling [19]. Lab-on-a-Chip (LoC) represents a family of devices which implement laboratory operations in a single chip of a few mm^2^–cm^2^ in size based on microfluidic technology, which allows precise flow control using different microchannel geometries [20,21]. In their microfluidic structure, these devices can execute miniaturized and automated operations routinely performed in laboratories, which permit quick testing of small molecules, including pathogen DNA, during food quality control [22,23]. This is allowed by microcapillary effects, based on the use of microchannels, where minimum quantities (a few microliters) of analytes are moved and mixed to obtain the desired reactions. Furthermore, capillary effects are orchestrated by the interaction between the surface tension of a liquid and the surface properties of its solid support [24]. LoCs, initially applied in the biomedical diagnostics, are now being introduced in the agri-food sector, to target food and environmental contaminants, mycotoxins, and allergens [23,25,26,27,28,29,30,31]. Moreover, different assays may be performed on a single instrument by selecting specific analytical protocols and using specific disposable cartridges.

The aim of the present study was to develop a rapid and cost-effective molecular analytical system based on LoC integrating real-time PCR or LAMP for specific detection of aflatoxigenic strains of *Aspergillus flavus* and *A. parasiticus* in hazelnut. The developed methods were compared to conventional molecular techniques in terms of sensitivity, specificity, speed, and repeatability. The developed system may be used as a platform to identify microbial contaminants in different food matrices by combining automation, portability, and cost-effectiveness of microfluidic technology with the high sensitivity of real-time PCR and LAMP. 

## 2. Results

### 2.1. Development of a Lab-on-a-Chip (LoC) Procedure

The experimental diagnostic equipment (RealLoC, LAMP srl, Scarmagno, Italy) used for LoC DNA amplification was developed based on a platform that includes three general controls: a thermal cycler, a fluorescence detector, and a firmware/software with the user interface (Figure 1A). Four LoCs were used in the system to simultaneously host and run the reaction. Two fluorescence emissions were included, which, by LED excitation at blue and green wavelengths, allowed the detection of 8 amplification curves per single run. Light emitted by the samples was collected and guided by an optical fiber system focused on the samples (Figure 1B). Fluorescence was chosen as a detection technique because of its reliability, specificity, and compliance with typical detection in laboratory diagnostic equipment. When amplification reactions occurred, blue and green light excitation LEDs caused emissions at every cycle (or, in the case of LAMP, every 30 s) from each LoC and wavelength.

Emitted radiation was collected and measured, producing sigmoidal-like curves as predicted by quantitative real-time PCR. Data collection was therefore elaborated by Sigmoidal Curve Fitting (SCF), second derivative calculated for each curve, and the zeroes of the functions were assumed as the Threshold Cycle (Ct) of the amplification reaction, according to commonly used techniques (Figure 2).

### 2.2. Development of a Lab-on-a-Chip (LoC)-LAMP and Loc-Real-Time PCR Assays

To determine and compare the performance of the LoC LAMP and the LoC real-time PCR, all the samples were first tested by standard LAMP and real-time PCR assays [17].

After successful amplification of aflatoxigenic *Aspergillus* spp. strains with the previously developed LAMP [17] and real-time PCR [32] methods, both LoC methods, LoC-LAMP and LoC-real-time PCR, were tested.

LoC-LAMP assay gave positive amplification signals for the three tested aflatoxigenic *Aspergillus* spp. strains. It showed an average detection of *A. flavus* from 23.29 to 106.6 min, and of *A. parasiticus* at 25.10 min, considering all DNA dilutions used (Table 1). The LoC-LAMP did not amplify non-aflatoxigenic *Aspergillus* spp. strains (*A. flavus* AF2, *Aspergillus sojae* BS8, *Aspergillus oryzae* AFCAL1, [17]). It also did not react with the negative control, consisting of water and the LAMP reaction mix.

The LoC-real-time PCR also allowed for the successful amplification of aflatoxigenic *A. flavus* strains, with average Ct values from 26.71 to 32.21 min. On the other hand, it was not able to amplify the aflatoxigenic *A. parasiticus* strain AFLX6. (Table 1). The LoC-real-time PCR did not amplify non-aflatoxigenic *Aspergillus* spp. strains (*A. flavus* AF2, *Aspergillus sojae* BS8, *Aspergillus oryzae* [17]) nor the negative control.

### 2.3. Specificity and Sensitivity of LoC-LAMP and LoC-Real-Time PCR

LoC-LAMP and LoC-real-time PCR were specific, as they yielded no positive results with non-toxigenic *Aspergillus* spp. strains (Table 1). In this way, both assays were non-cross-reactive and maintained the high specificity of the previously developed methods, LAMP and real-time PCR, from which the primers were used for the techniques developed in this study.

LoC-LAMP and LoC-real-time PCR assays showed sensitive assays for aflatoxigenic *A. flavus* strains. Thus, the LoC-LAMP amplified up to 10 fg of DNA and the LoC-real-time PCR up to 10 pg of DNA of *A. flavus* (Table 1). No satisfactory sensitivity results were obtained regarding *A. parasiticus*, as the LoC-real-time PCR did not produce any amplification with *A. parasiticus*, whereas the LoC-LAMP succeeded in amplifying 1 ng of fungal DNA.

### 2.4. Repeatability of LoC-LAMP and LoC-Real-Time PCR

Repeatability was evaluated by repeating the assays in three independent experiments with *A. flavus* AFSP4 and *A. parasiticus* AFLX6 strains with the same operator and instrument (RealLoC). Both assays proved to be repeatable due to the consistency of the amplification curves and the maximum shift of approx. 1 Ct or Tp (Table 2).

### 2.5. Stability of LoC-LAMP and LoC-Real-Time PCR

In the stability tests, the DNA extracts of the strains *A. flavus* AFSP4 and *A. parasiticus* AFLX6, as well as the reagents for the LoC tests, were stored at 4 °C for 7 days. The amplification profiles or detection limits of both species were not influenced by storage, and the amplification results were similar for the same samples tested at different storage times, 3 and 7 days, and did not exceed 5 Ct or Tp (Table 3).

## 3. Discussion

Standard molecular diagnostic methods are expensive, labor-intensive, and generally target only one pathogen, requiring highly specialized personnel and laboratory facilities. Moreover, they are poorly suitable for point-of-care analyses, and they are frequently long-lasting due to sample processing and test running. To overcome these limitations, new methodologies are required for sensitive and quick diagnosis directly in the field with portable, automatized, and miniaturized instruments.

Lab-on-a-Chip detection methods are quick and successful in disease biomarker identification and immunodetection [28,33,34,35]. LoC methods have been developed for the detection of *Aspergillus fumigatus*, a human pathogen, or for aflatoxigenic *Aspergillus* sp. in herbal species [36,37]. Microfluidic devices have also been developed for the detection of aflatoxins produced by *A. flavus* and *A. parasiticus*, such as a new biochip performing a competitive immunoassay for the detection of aflatoxin B1, a fiber microfluidic chip system enabling sensitive and rapid quantitative detection of AFB1, and paper-based analytical devices for rapid detection of AFB1, able to read colorimetric signals with smartphones [38,39,40].

In this study, previously developed primers for LAMP and real-time PCR techniques were used in miniature automated conditions of LoC and compared to the original techniques. Their successful use in LoC assays paves the way for an alternative miniature and microfluidic platform, which is cost-effective compared to standard molecular techniques. In detail, we succeeded in developing a specific, quick, and cheap assay based on microfluidic technique and fluorescence emission, capable of detecting 1.0 × 10^6^ cfu/mL of *A. flavus* in spiked hazelnut samples, through LoC real-time PCR and LoC LAMP techniques. On the other hand, *A. parasiticus* was successfully detected only with LoC LAMP assay at 1.0 × 10^7^ cfu/mL concentration, while it was not detected with LoC real-time PCR assay. The observed difference in detection limits between *A. flavus* (10 fg DNA) and *A. parasiticus* (1 ng DNA) confirms the genetic variability among these two species and accounts for a higher specificity of the original primers for *A. flavus*. Similar discrepancies were observed by Ortega et al. [17] with standard real-time PCR and LAMP assays. This indicates that LoC assays maintain the detection sensitivity of the original assays, but further optimization of the primer efficiency is necessary to achieve comparable detection limits between the two species. Although some of the technologies adopted in this study (e.g., standard polymer materials, routine micromilling, and ultrasonic welding procedures) are well documented in the literature, in this study, we used complex crude hazelnut extracts in LoC miniaturized cartridges, differently from most studies where LoC assays were developed with purified nucleic acids. The developed LoC, suitable for simplified crude DNA extraction from hazelnuts, opens the perspective of monitoring food-chain processes, avoiding the need for laborious and time-demanding DNA extraction [41].

In this study, the cost of the assay is importantly reduced. Thus, the LoC test material costs about EUR 1.5, making this assay economically interesting and competitive, particularly when compared with ELISA and real-time PCR, which cost ca. EUR 3.3 and EUR 2.7, respectively. A significant saving with respect to standard methods is also related to a minimal use of reaction reagents in the mix. Moreover, due to their simplicity, LoC tests can also be performed by minimally skilled personnel, an advantage for less advanced laboratories.

In this study, we successfully detected the target with miniaturized thermoplastic labs-on-a-chip (16 × 14 square millimeters), in line with previous studies [26,31,42], and no electrical detection is needed, so no microelectrodes were included in the LoC. A series of miniaturized and automated versions of the operations obtained by the instrument (named RealLoC) may permit point-of-care analyses and timesaving.

The detection limits found were 10 fg for LoC-LAMP and 10 pg for LoC-real-time PCR of *A. flavus*, which is similar to detection limits of other LoC-based methods for plant pathogens and food microorganisms [28,29,30,31], e.g., 14 fg for *Phytophthora* species [43], 10 fg for *Corynebacterium glutamicum* [44], and a few fg for *Escherichia coli* [45].

Altogether, the LoC developed in this study showed positive features, as decreasing costs, avoiding the use of commercial kits for DNA extraction, boosting simplicity and applicability in laboratory-limited conditions, and promoting a ‘sample-in-answer-out’ testing strategy. The developed test could be combined with the detection of other hazelnut pathogens, which would further decrease the costs and permit the multiplexing of the assay. Future inter-laboratory validation tests are needed to confirm the robustness of the developed method.

## 4. Conclusions

Overall, the LoC assays developed in this study provide a miniaturized, effective, and rapid way to identify aflatoxigenic *Aspergillus flavus* and *A. parasiticus*; however, a few potential difficulties and challenges should be taken into consideration. The lower sensitivity and slower detection time of the developed LoC assays compared to standard real-time PCR and LAMP methods require future improvements to the geometric structure or materials in the microfluidic chip, which will contribute to the improvement of sensitivity and speed of results while maintaining cost savings.

The complexity of the LoC techniques developed in this study could be further simplified through the development of pre-prepared and ready-to-use kits, and automated data interpretation in order to facilitate their use by technical personnel without advanced skills in molecular biology. The RealLoC instrument should be adapted and equipped with additional batteries for direct point-of-care use in the field, as well as during nut storage and processing. Another constraint of the LoC is the low number of samples that can be processed in a single assay; therefore, it is necessary to increase the number of reactions per assay to increase its practicability. Currently, LoC technologies are not uniform or harmonized, particularly in terms of the materials used to build the chips. When a higher degree of harmonization in the materials used is reached, LoC will be more suitable for high-throughput and user-friendly detection, which will enable better entry into the market, as already seen with LoCs for HIV detection, glucose monitoring, or heart attack diagnostics [46,47,48].

In conclusion, a developed LoC assay paves the way for alternative monitoring of aflatoxigenic fungi and quality control in the agri-food industry with a point-of-care analysis, free from commercial kit DNA extraction.

## 5. Materials and Methods

### 5.1. Fungal Isolates

Two isolates of *A. flavus* (AFSP4 and AFLX8) and one isolate of *A. parasiticus* (AFLX6), capable of producing aflatoxins, isolated from hazelnut [17], were used in this study. Additionally, two non-aflatoxigenic strains (*A. flavus* FS6 and *A. parasiticus* FV4) were used. The isolates were grown on Yeast Extract Sucrose agar (YES) [49] at 30 °C for one week. The conidia were collected by scraping the mycelium with a loop in a sterile 1% Tween 20 solution. The conidial suspension was taken from a Petri dish, and its concentration was determined by using a Bürker chamber. The final concentration was adjusted to 1 × 10^5^, 1 × 10^6^, and 1 × 10^7^ conidia/mL.

### 5.2. Inoculation of Hazelnuts

Freshly harvested hazelnuts were stored in sealed polypropylene plastic bags at 4 °C before the experiment initiation. Hazelnuts were sterilized with 1% sodium hypochlorite, washed with sterile deionized water, and allowed to air dry. Three wounds were made per hazelnut (1 cm long), and 500 µL of the fungal spore suspension was inoculated in each wound at three different concentrations (1 × 10^5^, 1 × 10^6^, 1 × 10^7^ conidia/mL). Control nuts were inoculated with sterile deionized water. After inoculation, hazelnuts were stored in perforated plastic bags in three replicates at room temperature for one week. The experiment was carried out twice.

### 5.3. DNA Extraction

A crude DNA extraction was performed using an alkaline extraction protocol with slight modifications as described by Chomczynski and Rymaszewski [50]. One hazelnut was placed in 5 mL tubes containing 1 mL of the crude extraction PEG buffer (50 g/L 4600 PEG; 20 mmol/L KOH; pH 13.5) and homogenized by vigorous manual shaking for 1–3 min. The homogenates were diluted to 1:50 in sterile water, reaching 500 μL of the final volume.

The quality of DNA was checked using a NanoDrop 2000 Spectrophotometer (Thermo Scientific, Wilmington, DE, USA). DNA concentration was calculated using the conversion factor of 50 ng/μL for 1 optical density unit at 260 nm [51], and the DNA was diluted using ten-fold serial dilutions ranging from 1 ng to 1 fg/µL.

Finally, diluted crude extracts were added to the isothermal or real-time amplification mixture as described below. The same set of samples was used for all four molecular assays (LAMP, LoC-LAMP, real-time PCR, and LoC-real-time PCR).

### 5.4. Real-Time PCR and LAMP

A LAMP assay was carried out following the protocol described by Ortega et al. [17]. Each 12 µL LAMP reaction contained 1 μL DNA, 2.3 μM FIP and BIP, 0.3 μM F3 and B3, and 1.2 μM FL and BL primers and 6 µL Isothermal Master Mix (ISO-004, OptiGene, Horsham, UK). The LAMP reaction was performed at 65 °C for 60 min using the StepOnePlus Real Time PCR (Applied Biosystems, Loughborough, UK).

A real-time PCR assay was performed using the protocol described by Cruz and Buttner [32] with slight modifications. Amplification was carried out in 96-well Optical Reaction Plates (Applied Biosystems) sealed with a MicroAmp^TM^ Optical Adhesive cover using a StepOnePlus Real-Time PCR (Applied Biosystems, Loughborough, UK). The 25 µL real-time PCR mixture contained Power SYBR^®^ Green PCR Master Mix (Applied Biosystems, Loughborough, UK), 0.2 μM of each primer (Asp1S and AflR2), and 1 µL DNA. The cycling conditions included an incubation at 95 °C for 10 min, followed by 40 cycles of 10 s at 95 °C and 30 s at 60 °C, carried out in the StepOnePlus Real Time PCR (Applied Biosystems, Loughborough, UK).

### 5.5. LoC Fabrication

In the literature, several materials were considered for the fabrication of microfluidic cartridges, including silicon, glass, and polymers. Due to their optimal optical and inertial performances, low cost, and simple manufacturing, after some preliminary tests, polymers were selected for the fabrication of the LoC cartridges.

Among the tested polymers, including polymethylmethacrylate (PMMA), polycarbonate (PC), polypropylene (PP), olefins, and cycloolefin copolymer (COC, TOPAS^®^, TOPAS Advanced Polymers GmbH, Tokyo, Japan), COC was finally selected as the best candidate due to its excellent transparency, low autofluorescence, good biocompatibility, and optimal manufacturing process.

After the definition of the diagnostic protocol (cycle temperatures, detection method, sample amount, reagents’ chemical composition, etc.), and some thermal behavior simulations, different microfluidic geometries of the Lab-on-Chip were designed.

The micromilling technique was first employed to realize a few samples of microchannel structures, with different geometries (Figure 3A,B): They have in common insertion bores for liquid introduction and a microchannel structure (500 micron wide). In Figure 3B, the microchamber (about 8–12 microliters), which is useful for efficiently performing the amplification reactions, is highlighted. The overall surface area of the LoC is about 14 × 16 mm^2^.

An Ultrasound welding machine (RINCO ULTRASONICS AG, Romanshorn, Switzerland) was used to seal the micro-milled prototype cartridges and execute the prime experiments. At the end of this preliminary phase, a mold for the injection was created: each of the two locations had four different geometries in order to achieve more alternatives for subsequent diagnostic tests (Figure 4A, injection mold).

Finally, a large number (about a thousand) of microfluidics were realized by the injection molding technique (Figure 4B, molded microfluidics, still with sprue), singularized and sealed with the same COC polymer plain cover by a US welding machine. Thermal treatments were performed before executing the tests to ensure stability, cleaning, and sterilization of the COC cartridges.

### 5.6. LAMP LoC

A LAMP was carried out in the RealLoC instrument developed in-house by LAMP srl (see Figure 1). Each 12 µL reaction contained 1 μL DNA, 2.3 μM FIP and BIP, 0.3 μM F3 and B3, and 1.2 μM FL and BL primers and 6 µL Isothermal Master Mix (ISO-004, OptiGene, Horsham, UK). The cycling conditions included 60 min at 65 °C, as those described for LAMP.

### 5.7. Real-Time LoC

A real-time PCR was also carried out in the RealLoC diagnostic instrument (LAMP srl). The 12 µL real-time PCR mixture contained 6 µL Power SYBR^®^ Green PCR Master Mix (Applied Biosystems), 0.2 μM of each primer (Asp1S and AflR2), and 1 µL DNA. The cycling conditions were the same as those described for the real-time PCR.

## Figures and Tables

**Figure 1 toxins-17-00510-f001:**
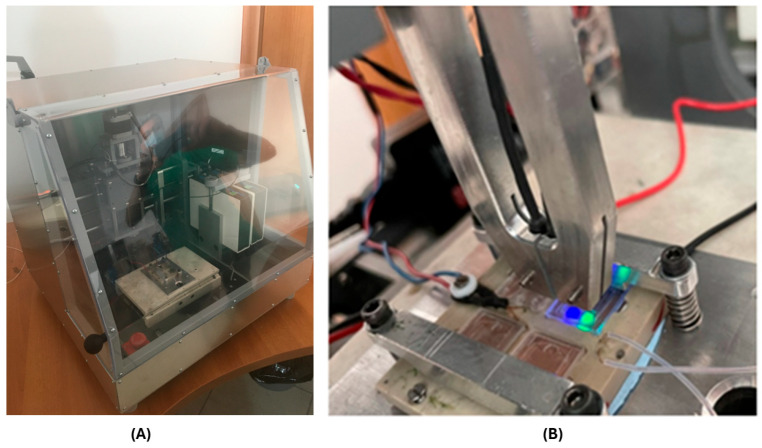
(**A**) The RealLoC instrument. On the left, the Peltier-based thermocycler, where the four LoCs are placed to be processed. A laptop pc runs a dedicated app to perform the tests, collect data, and elaborate results. (**B**) A particular aspect of the optical fiber detection system, with the two excitation LEDs switched on. The moving arm mounts the two excitation/collecting fluorescence optical fibers, which provide, respectively, blue and green excitation light from the LED sources. The same fibers can collect emissions from the four samples and bring them to the two sensors (visible in Figure 1A, to the right of the instrument).

**Figure 2 toxins-17-00510-f002:**
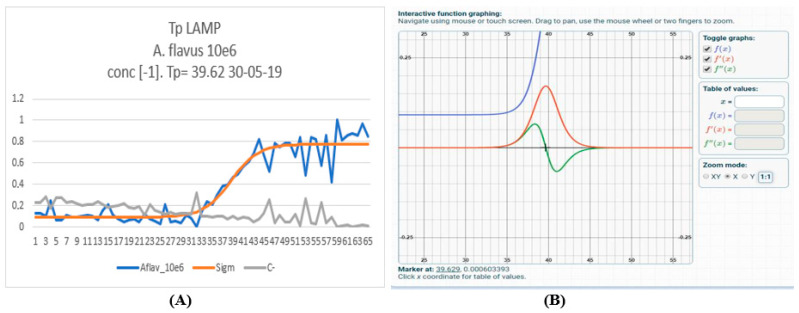
(**A**) Specific detection of aflatoxigenic *Aspergillus flavus* by LoC-LAMP assay using the RealLoC instrument. Typical data (blue) from an amplification reaction generated by the RealLoC instrument, and their fitting by a sigmoidal curve (orange). (**B**) First and second derivatives of Sigmoidal Curve Fitting (SCF), highlighting flexus position.

**Figure 3 toxins-17-00510-f003:**
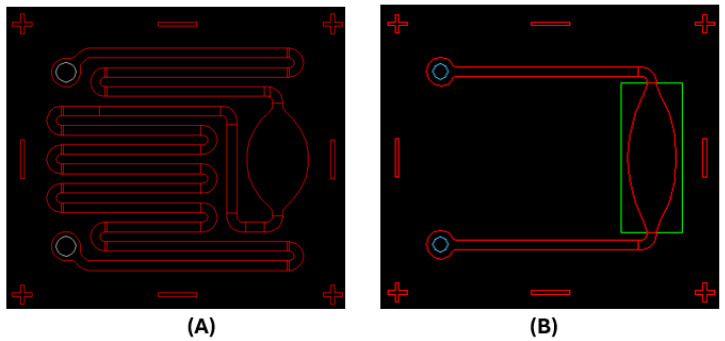
Two examples of simple microfluidics for LoC: (**A**) long microchannels and a reaction microchamber; (**B**) short channel sleeves and a microchamber. Both include in/out microbores.

**Figure 4 toxins-17-00510-f004:**
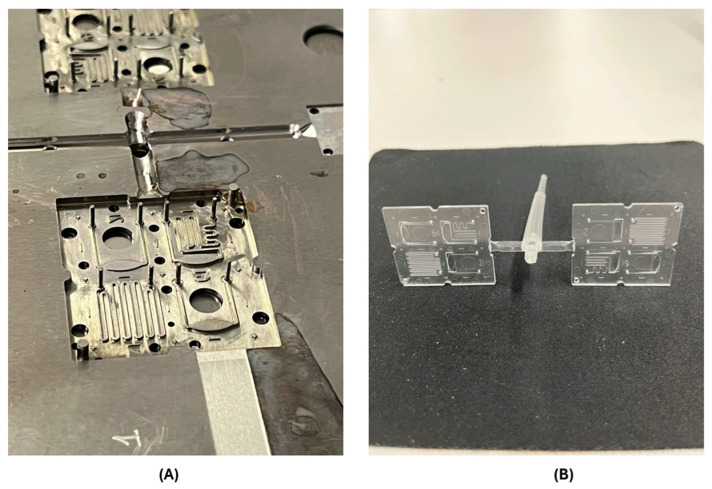
(**A**) The mold realized for LoC injection molding (single site shown). (**B**) Double-site microfluidics, each including different geometries.

**Table 1 toxins-17-00510-t001:** Comparison of the sensitivity and specificity of four diagnostic techniques for specific detection of *Aspergillus flavus* and *A. parasiticus:* LAMP, real-time PCR, LoC-LAMP, and LoC-real-time PCR.

Species	DNA Concentration	LoC-LAMP	LoC-Real-Time PCR	LAMP	Real-Time PCR
		Tp * (min:s)	Ct **	Tp (min:s)	Ct
*A. flavus* AFSP4	1 ng	19:02	28.41 ± 1.83	23:99 ± 0.04	6.92 ± 0.27
*A. flavus* AFLX8		25:75	25.00 ± 1.75	20:66 ± 1.77	7.08 ± 0.23
*A. flavus* FS6		nd ***	nd	nd	nd
*A. parasiticus* AFLX6		25:10	nd	20:52 ± 0.62	7.04 ± 0.16
*A. parasiticus* FV4		nd	nd	nd	nd
*A. flavus* AFSP4	10 pg	39:62	34.15 ± 0.22	26:85 ± 0.01	10.68 ± 0.34
*A. flavus* AFLX8		63:08	30.27 ± 0.17	26:97 ± 0.16	10.72 ± 0.50
*A. flavus* FS6		nd	nd	nd	nd
*A. parasiticus* AFLX6		nd	nd	24:95 ± 1.25	10.64 ± 0.39
*A. parasiticus* FV4		nd	nd	nd	nd
*A. flavus* AFSP4	10 fg	106:6	nd	29:38 ± 0.00	14.74 ± 0.33
*A. flavus* AFLX8		nd	nd	30:85 ± 0.31	15.58 ± 0.56
*A. flavus* FS6		nd	nd	nd	nd
*A. parasiticus* AFLX6		nd	nd	31:31 ± 0.96	15.07 ± 0.15
*A. parasiticus* FV4		nd	nd	nd	nd
*A. flavus* AFSP4	1 fg	nd	nd	35:61 ± 0.31	26.46 ± 0.30
*A. flavus* AFLX8		nd	nd	35:01 ± 0.29	27.63 ± 0.13
*A. flavus* FS6		nd	nd	nd	nd
*A. parasiticus* AFLX6		nd	nd	36:94 ± 0.40	28.11 ± 0.52
*A. parasiticus* FV4		nd	nd	nd	nd

* Tp = time to positive, ** Ct = threshold cycle, *** nd = not determined.

**Table 2 toxins-17-00510-t002:** Repeatability of LoC-LAMP and LoC-real-time PCR for specific detection of *Aspergillus flavus* and *A. parasiticus*. Values were expressed as threshold cycle (LoC-LAMP) and time to positive (LoC real-time PCR).

Species	LoC-LAMP	LoC-Real-Time PCR	LoC-LAMP	LoC-Real-Time PCR
	1 ng		10 pg	
*A. flavus*	19.071 ± 0.88	28.88 ± 1.76	40.74 ± 1.07	34.20 ± 0.31
*A. parasiticus*	24.86 ± 0.56	nd *	nd	nd

* nd = not determined.

**Table 3 toxins-17-00510-t003:** Stability of LoC-LAMP and LoC-real-time PCR for specific detection of *Aspergillus flavus* and *A. parasiticus*. Values were expressed as threshold cycle (LoC-LAMP) and time to positive (LoC real-time PCR).

Species	LoC-LAMP		LoC-Real-Time PCR		LoC-LAMP		LoC-Real-Time PCR	
	1 ng				10 pg			
	3 d *	7 d	3 d	7 d	3 d	7 d	3 d	7 d
*A. flavus*	20.01 ± 0.43	22.25 ± 0.68	28.70 ± 1.12	30.39 ± 1.89	42.74 ± 1.51	44.14 ± 2.04	34.50 ± 0.78	35.37 ± 0.89
*A. parasiticus*	24.53 ± 0.77	24.90 ± 1.26	nd **	nd	nd	nd	nd	nd

* d = day, ** nd = not determined.

## Data Availability

The original contributions presented in this study are included in the article. Further inquiries can be directed to the corresponding author.

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
