# Peer review of "Development of Lab-on-a-Chip LAMP and Real-Time PCR Assays to Detect Aflatoxigenic Aspergillus flavus and Aspergillus parasiticus in Hazelnuts"

_toxins, 2025, doi:10.3390/toxins17100510_

Round 1
Reviewer 1 Report (Previous Reviewer 2)
Comments and Suggestions for Authors
After careful evaluation, I recommend that this manuscript can be considered for acceptance after minor revisions. However, there are a few issues that should be addressed to improve the overall quality of the paper:
-
The English expression throughout the manuscript could be further polished for better readability and fluency, particularly in the Discussion section.
-
The references section requires attention to ensure consistency in formatting. Please adhere strictly to the journal’s guidelines.
-
The clarity of Figure 4 needs improvement. Higher resolution or better labeling would enhance its interpretability.
Author Response
- The English expression throughout the manuscript could be further polished for better readability and fluency, particularly in the Discussion section.
English throughout the manuscript was improved, thanks to the accurate revision of an English mother tongue reviewer.
- The references section requires attention to ensure consistency in formatting. Please adhere strictly to the journal’s guidelines.
The refrences were carefully checked and corrected following the Journal’s guidelines.
- The clarity of Figure 4 needs improvement. Higher resolution or better labeling would enhance its interpretability.
Figure 4 was improved as requested by the reviewer.
Reviewer 2 Report (New Reviewer)
Comments and Suggestions for Authors
The study demonstrates the application of a lab-on-a-chip (LoC) platform for the detection of aflatoxigenic Aspergillus flavus and A. parasiticus. While the subject matter is relevant, the manuscript's originality and scientific contribution are limited for several reasons.
(1) The study is based entirely on previously published primers and protocols (Ortega et al., 2020; Cruz & Buttner, 2008). The absence of novel primer design or assay optimization detracts from the innovation of the molecular component.
(2) The microfluidic cartridge is fabricated using standard polymer materials (COC) and conventional micromilling and ultrasonic welding methods. These techniques are well established in the literature, yet the manuscript does not clearly articulate how the present LoC design improves upon or differs from existing LoC approaches.
(3) While the authors emphasize reduced reagent consumption, low cost, and simplified handling, the data on sensitivity and reproducibility is inconsistent. The detection limits differ substantially between A. flavus (10 fg DNA by LoC-LAMP) and A. parasiticus (1 ng DNA). This discrepancy raises questions about the robustness and general applicability of the method. Additionally, the system's inability to detect certain toxins (e.g., AFB1) restricts its effectiveness in food safety monitoring.
(4) The discussion places significant emphasis on cost-effectiveness and reduced reagent requirements; however, it does not adequately contextualize this work within the broader framework of other LoC-based detection systems for foodborne fungi. The manuscript would benefit from a more critical assessment of its true innovation and how it advances beyond prior studies.
While the application of LoC to hazelnut monitoring is noteworthy, the manuscript would benefit from clearer articulation of its novelty, stronger validation data, and a more rigorous discussion of how it differs from and improves upon existing LoC-based molecular diagnostics.
Author Response
- The study is based entirely on previously published primers and protocols (Ortega et al., 2020; Cruz & Buttner, 2008). The absence of novel primer design or assay optimization detracts from the innovation of the molecular component.
The novelty of our study was not to develop new primers but to all study to use of the same primers in newly developed miniaturized and automated LoC techniques which offer valid alternative for detection of of aflatoxigenic A. flavus and A. parasiticus. To more emphasize this point we added in revised manuscript this sentence ‘In this study, previously developed primers for LAMP and real-time PCR techniques were used in miniature automated conditions of LoC and compared to the original tech-niques. Their successful use in LoC assays paves the way for an alternative miniature and microfluidic platform which is cost effective compared to standard molecular techniques’.
- The microfluidic cartridge is fabricated using standard polymer materials (COC) and conventional micromilling and ultrasonic welding methods. These techniques are well established in the literature, yet the manuscript does not clearly articulate how the present LoC design improves upon or differs from existing LoC approaches.
Although standard polymer materials, conventional micromilling and ultrasonic welding are well described in the literature, in this study we used crude hazelnut extracts in LoC miniaturized catridges which are complex extracts that present the novelty compared to previously developed LoC assays that are primarily used in medicinal field in combination with purified nucleic acids. To more emphasize that point we added this sentence to the discussion.
- While the authors emphasize reduced reagent consumption, low cost, and simplified handling, the data on sensitivity and reproducibility is inconsistent. The detection limits differ substantially between A. flavus (10 fg DNA by LoC-LAMP) and A. parasiticus (1 ng DNA). This discrepancy raises questions about the robustness and general applicability of the method. Additionally, the system's inability to detect certain toxins (e.g., AFB1) restricts its effectiveness in food safety monitoring.
The observed difference in detection limits between A. flavus (10 fg DNA) and A. parasiticus (1 ng DNA) confirms reported genetic variability among these two species. Similar discrepancies were observed in Ortega et al. 2020 study by use of routine PCR and LAMP assays. This indicates that LoC assays maintain detection sensitivity, while further optimization (e.g. primer efficiency) is required to achieve comparable detection limits between these two species. This part is also added to revised manuscript.
- The discussion places significant emphasis on cost-effectiveness and reduced reagent requirements; however, it does not adequately contextualize this work within the broader framework of other LoC-based detection systems for foodborne fungi. The manuscript would benefit from a more critical assessment of its true innovation and how it advances beyond prior studies. While the application of LoC to hazelnut monitoring is noteworthy, the manuscript would benefit from clearer articulation of its novelty, stronger validation data, and a more rigorous discussion of how it differs from and improves upon existing LoC-based molecular diagnostics.
We tried to improve the novelty and validation data in the revised manuscript of the article, as well as the discussion part.
Reviewer 3 Report (Previous Reviewer 3)
Comments and Suggestions for Authors
This is a revision of the previous submitted manuscript, but further revision is still needed.
1) This is a research paper, so result part 2.3 shoul be enlarged, the Specificity, sensitivity, reproducibility and stability are very important. The author should rearranage this part, such as 2.3 Specificity and sensitivity, 2.4 reproducibility and 2.6 stability. Every part needs data or tabe or figure to support itself.
Author Response
- This is a research paper, so result part 2.3 shoul be enlarged, the Specificity, sensitivity, reproducibility and stability are very important. The author should rearranage this part, such as 2.3 Specificity and sensitivity, 2.4 reproducibility and 2.6 stability. Every part needs data or tabe or figure to support itself.
We separated the specificity, sensitivity, repeatability and stability in separate sections in the revised version of the manuscript as required by the reviewer. The data for each section was presented (specificity and sensitivity in Table 1) and repeatability and stability were presented in Figures 3 and 4 (which were added to the revised manuscript).
Round 2
Reviewer 2 Report (New Reviewer)
Comments and Suggestions for Authors I am in agreement with the acceptance of this paper as the authors have adequately addressed the concerns of my peer review.This manuscript is a resubmission of an earlier submission. The following is a list of the peer review reports and author responses from that submission.
Round 1
Reviewer 1 Report
Comments and Suggestions for Authors
The topic is very interesting and within the scope of this Journal. Maybe you can take these comments into consideration. I'm very sorry, but I believe there is no planned and structured methodology. I also don't see a robust method validation work. The conclusion of the article has not been robust and convincing enough to justify the study and its applicability
Line 18: I think the word “mycotoxin” should be added.
Introduction
Line 22: A comma should be added “persistence, and toxin”.
Line 25: “monitor” instead of “monitors”.
Line 28: The word “The” before “food” should be omitted.
Line 31: Sorry, I don´t understand “link” in this sentence. I don´t understand why aflatoxins G1 and G2 are not included.
Line 70: “environmental contaminants” instead of “environmental pollutants”.
Line 77: “real-time PCR” instead of “real tim PCR”.
Results
Line 84 and 85. A. parasiticus should be written in italics.
Table 1.- Number of spores / ml-1 Maybe you have to change to spores/ml. It is not indicated in the table what "nt" means too.
Line 111: Figures 1A and 1B do not provide additional information
Materials and methods
Line 195: “Yeast extract Sucrose agar (YES)”, Extract should be write in capital letters.
Line 196: The word “solution“ should be added. “loop in a sterile 1% Tween 20 solution”.
Line 219: EZNA kit should be explained; I mean, what is it used for?. Supplier should be added too.
Line 229: For example, the city and location are not included. “PCR Master Mix (Applied Biosystems), ”For example, in the line 232 is included. Please, check it.
Line 253: Figure 3A is not included in the text like for example 3B.
Reviewer 2 Report
Comments and Suggestions for Authors
This study developed a lab-on-a-chip assay based on LAMP for detecting Aspergillus flavus and Aspergillus parasiticus in hazelnuts. The manuscript is acceptable in terms of technical aspects. However, the manuscript requires major revision before further consideration. To improve the manuscript, I recommend the authors address the following points:
1. The abstract part includes too much background information, and please add some necessary result data.
2. Introduction. The aim of the manuscript should be explained better in the introduction.
3. Please add the mechanism of this method for detecting Aspergillus flavus and Aspergillus parasiticus.
4. Please add the stability experiment and feasibility of the LAMP experiment.
5. Results and Discussion. Authors are advised to discuss and, if possible, compare their results with those of other similar studies and add corresponding references.
6. Please add the "Conclusion" part. This part need to discuss the potential limitations and challenges of the proposed method.
7. References. Authors should cite the recent five years of literature.
Reviewer 3 Report
Comments and Suggestions for Authors
In this study, the authors developed a lab-on-a-chip method based on real-time PCR and on LAMP for the specific detection of toxigenic A. flavus and A. parasiticus from infected hazelnuts. The microfluidic chip allowed a quick, specific, sensitive, simple, automatized, cheap, and us-er-friendly A. flavus and A. parasiticus detection.The miniaturized size of the lab-on-a-chip is more suitable for the native microenvironment of aflatoxins, which allows a more reliable fungal detection. It has significance for the readers in these fields. My suggestion is acceptable after minor revision. Other comments were followed.
1) There are many methods for A. flavus detection, why the author select this mehtod? is it sensitive? timely? low-costly? please confirm in the discussion part.
2) The author used A. flavus and A. parasiticus for detection. but why not add other species, such as A. nidulans, A. oryzae or A. sojae to show its specificity?
3) A. flavus can be detected using this developmented method, but how about its toxins such as AFB1?
4) 2.2 should be first display for its pricinple, then giving its detection results in 2.1. so you should change 2.2 to 2.1, and change 2.1 to 2.2.
5) After you obtained this method, where and which fields will be used or needed for this method?
6) English language need improvement.
Comments on the Quality of English LanguageThe English language need a moderate editing.
Reviewer 4 Report
Comments and Suggestions for Authors
This work developed a lab-on-a-chip method based on real-time PCR and LAMP for the detection of toxigenic A. flavus and A. parasiticus from infected hazelnuts. However, the author should provide clearer explanations and more detailed data to demonstrate the conclusions. Therefore, I think that major revisions need to be made to the paper before reconsidering its publication.
1. Are primers and other reagents used in this work all from commercial reagent kits? The sensing strategy design of this work only combines traditional real-time PCR and LAMP with lab-on-a-chip, so what are the main innovations of this sensing strategy?
2. The experimental data provided by the author is relatively limited. Please provide sufficient data and illustrate the main performance such as sensitivity, specificity, speed, and repeatability in the form of figures or tables.
3.The Figure 2 provided by the author is too blurry. Please provide a clear image to facilitate readers' in-depth understanding of the work.
Comments on the Quality of English LanguageMinor editing of English language required.